# FILLING IN THE DETAILS:
# PERCEIVING FROM LOW FIDELITY VISUAL INPUT

**Farahnaz A. Wick**[*]
University of Massachusetts Boston,
Boston, MA 02125
fwick@cs.umb.edu

**Michael L. Wick**
University of Massachusetts Amherst,
Amherst, MA 01003
mwick@cs.umass.edu

**Marc Pomplun**
University of Massachusetts Boston,
Boston, MA 02125
mpomplun@cs.umb.edu

## ABSTRACT

Humans perceive their surroundings in great detail even though most of our visual field is reduced to low-fidelity color-deprived (e.g., dichromatic) input by the retina. In contrast, most deep learning architectures deploy computational resources homogeneously to every part of the visual input. Is such a prodigal deployment of resources necessary? In this paper, we present a framework for investigating the extent to which connectionist architectures can perceive an image in full detail even when presented with low acuity, distorted input. Our goal is to initiate investigations that will be fruitful both for engineering better networks and also for eventually testing hypotheses on the neural mechanisms responsible for our own visual system's ability to perceive missing information. We find that networks can compensate for low acuity input by learning global feature functions that allow the network to fill in some of the missing details. For example, the networks accurately perceive shape and color in the periphery, even when 75% of the input is achromatic and low resolution. On the other hand, the network is prone to similar mistakes as humans; for example, when presented with a fully grayscale landscape image, it perceives the sky as blue when the sky is actually a red sunset.

## 1 INTRODUCTION

Most deep learning architectures process every visual input component when performing a task; for example, the input layer of many ImageNet architectures considers all pixels in every region of a pre-processed image when learning an image classifier or making classification decisions (Krizhevsky et al., 2012; Simonyan & Zisserman, 2014; Szegedy et al., 2014). In contrast, the human visual system has just a small fovea of high resolution chromatic input allowing it to more judiciously budget computational resources (Lennie, 2003). In order to receive additional information in the field of view, we make either covert or overt shifts of attention. Overt shifts of attention or *eye-movements* allow us to bring the fovea over particular locations in the environment that are relevant to current behavior. To avoid the serial nature of processing as demanded from overt shifts of attention, our visual system can also engage in covert shifts of attention in which the eyes remain fixated on one location but attention is deployed to a different location.

Yet, even without resorting to overt shifts of attention, we still perceive the world in high detail. This is somewhat remarkable if you consider that the human retina receives an estimated 10 million bits per second which far exceeds the computational resources available to our visual system to assimilate at any given time (Koch et al., 2006). Our own fovea takes up only 4% of the entire retina (Michels & CP Rice, 1990) and is solely responsible for sharp central full color vision with maximum acuity; acuity which diminishes rapidly with eccentricity from the fovea (Cowey &

---

[*]Email: fwick@cs.umb.edu

Rolls, 1974). As a result, visual performance is best at the fovea and progressively worse towards the periphery (Low, 1946). Indeed, our visual cortex is receiving distorted color-deprived visual input except for the central two degrees of the visual field (Hansen et al., 2009). Additionally, we have blind spots in the retina that receive no visual input. Yet, we are mostly unaware of these distortions. Even when confronted with actual blurry or distorted visual input, our visual system is good at extracting the scene contents and context. For instance, our system can recognize faces and emotions expressed by those faces in resolutions as low as 16 x 16 pixels (Sinha et al., 2006). We can reliably extract contents of a scene from the gist of an image (Oliva, 2005) even at low resolutions (Potter & Levy, 1969; Judd et al., 2011). Recently, Ullman et al. (Ullman et al., 2016) has shown that our visual system is capable of recognizing contents of images from critical feature configurations (called minimal recognizable images or MIRCs) that current deep learning systems cannot utilize for similar tasks. These MIRCS resemble foveations on an image and their results reveal that the human visual system employs features and processes that are not used by current deep networks. Similarly, little attention has been given to how these networks deal with distorted or noisy inputs. We draw inspiration from the abilities of the human visual system and propose a framework to study questions related to whether an artificial neural network can learn to perceive an image from low fidelity input.

In this paper, we want to understand what kind of information can be gleaned from low-fidelity inputs. What can be gleaned from a single foveal glimpse? What is the most predictive region of an image? We present a framework for studying such questions based on autoencoders. In contrast to traditional or denoising autoencoders (Vincent et al., 2008), which learn to reconstruct the original image in the presence of random salt and pepper noise, our autoencoders attempt to reconstruct original high-detail inputs from systematically corrupted lower-detail foveated versions of those images (that is, images that are entirely low detail except perhaps a small "fovea" of high detail). Thus, we have taken to calling them defoveating autoencoders (DFAE). We find that even relatively simple DFAE architectures are able to perceive color, shape and contrast information, but fail to recover high-frequency information (e.g., textures) when confronted with extremely impoverished input. Interestingly, as the amount of detail present in the input diminishes, the structure of the learnt features becomes increasingly global.

## 2 RELATED WORK

Corrupting the input or hidden layers via noise to improve neural networks is an area of active study (Bishop, 1995; LeCun et al., 1989; Vincent et al., 2010; Rifai et al., 2011; Xie et al., 2012; Schuler et al., 2013). A highly related framework is denoising autoencoders in which the input (or sometimes hidden layer) is corrupted and the network must reproduce the non-corrupted output (Vincent et al., 2010). However, the form of our input corruption is systematic, not random. Further, the emphasis of our work is to understand to what extent a given architecture can perform *perceptual filling-in* (Komatsu, 2006), our brain's ability to perceive content that is not explicitly present in our visual input, from retina-like distorted inputs. We study this capability by varying the type of input distortion and examining (qualitatively and quantitatively) the ability of the network to perceive.

Note that one can consider a low-resolution image as yet another type of noisy input. Thus, another area of related work is image super-resolution in which the goal is to learn a transform from a low-resolution image to a high-resolution image (Behnke, 2001; Cui et al., 2014; Dong et al., 2014), and also image denoising (Jain & Seung, 2009). These are exciting applications for deep learning, but again, our emphasis is on studying a specific set of scientific questions rather than engineering a solution to a specific image-cleaning problem.

There has been tremendous interest in applying attention to deep learning architectures (Larochelle & Hinton, 2010; Mnih et al., 2014; Bahdanau et al., 2014; Xu et al., 2015). Such work has lead to improvements in tasks ranging from machine translation (Bahdanau et al., 2014) to image captioning (Xu et al., 2015). In many of these frameworks, attention interacts with the limited resources in various ways (either by sequentially directing attention (Mnih et al., 2014) or by allowing the network to recall relevant information from the past history (Xu et al., 2015)). Our work is complementary in that we aim to study the limited resource itself: in this work, a single foveal glimpse. We further suggest a way of extending our framework to incorporate attention, but we save investigation for future work.

## 3 FRAMEWORK: DEFOVEATING AUTOENCODERS (DFAE)

We now present a framework for studying the extent to which neural networks can "perceive" an image given various types of low-detail (or foveated) inputs. We begin by specifying a space of neural network architectures and by precisely defining a notion of *perceives* that we can measure. It is important that the framework is general and not dependent on a specific task such as image classification in which, for example, the ability to learn domain-specific discriminating features might make it easy to solve the classification problem without fully modeling the structure of the input. This is undesirable because then we are unable to trust classification accuracy as a reliable surrogate for *perceiving*.

With this in mind, we focus instead on generative models of the raw input data itself, specifically autoencoders (AE). The AE's hidden units $h$ are analogous to the intermediate neurons in our visual system that capture features and structure of the visual input. Similarly, the AE's weights $W$ forge visual memories of the training set and are thus analogous to long-term memory. When these weights are properly trained, the activations of the hidden units reflect how the network is perceiving a novel input. However, since these units are not directly interpretable, we indirectly measure how well the network perceives by evaluating the similarity between the original and generated (high-detail) images: the more similar the images are, the better the network is able to perceive.

More formally, let $x$ be the original input image and $\hat{x} = \phi(x)$ be a lower-detail *foveated* version of that image. That is, a version of the image which is mostly low-detail (e.g., downsampled, black-and-white, or both) except for possibly a small portion which is high-detail (mimicking our own fovea). For example, if we encode images as vectors of floats between 0 and 1 (reflecting pixel intensities in RGB or grayscale) then we might define a class of foveation functions as $\phi : [0,1]^n \to [0,1]^m$ s.t. $m \ll n$ and the foveation function might downsample the original image according to the eccentricity from the image center while also removing most of the vector components corresponding to color. We then employ the autoencoder to defoveate $\hat{x}$ by generating a high-quality output image $y = f(\hat{x}; W)$ in which, for example, $y \in [0,1]^n$. Finally, we can then measure the similarity between $y$ and $x$ as (1) a surrogate for how well the network perceives from the foveated input and (2) part of a loss function to train the network.

In summary, DFAEs simply comprise:

1. A foveation function that filters an original image by removing detail (color, downsampling, blurring, etc). We will later make this the independent variables in our experiments so we can study the effect of different types of input distortion.

2. An autoencoder network that inputs the low-detail foveated input, but is trained to output the high-detail original image.

3. A loss function for measuring the quality of the reconstruction against the original image and for training the network.

Note that much like denoising autoencoders, these autoencoders reconstruct an original image from a corrupted version. However, the form of corruption is a systematic foveation instead of random noise (Vincent et al., 2008). Thus, we have termed these models *defoveating* autoencoders or DFAEs.

### 3.1 DFAE ARCHITECTURE AND LOSS FUNCTION

In our experiments, we study DFAEs with fully connected layers. That is, DFAEs of the form:

$$\hat{x} = \phi(x) \qquad\qquad\qquad h^{(0)} = \tanh\left(W^{(0)}\hat{x}\right)$$

$$h^{(i)} = \tanh\left(W^{(i)}h^{(i-1)}\right) \quad \text{for } i = 1, \cdots, k-1 \qquad y = \sigma\left(W^{(k)}h^{(k-1)}\right)$$

where $\sigma$ is the logistic function. The sigmoid in the final layer conveniently allows us to compare the pixel intensities between the generated image $y$ and the original image $x$ directly, without having to post-process the output values. We experiment with the number of hidden units per-layer as well as the number of layers. For training, we could employ the traditional mean-squared (MSE) error or cross-entropy loss, but we found that the domain-specific loss function of peak signal-to-noise ratio

(PSNR) yielded much better training behavior. The PSNR between generated image a $y = f(\phi(x))$ and its original input $x$ is defined as follows:

$$L_H(x, y) = \log_{10}\left(\frac{1}{\sqrt{MSE(x, y)}}\right) \quad \text{where } MSE(x, y) = n^{-1}(x - y)^T(x - y) \quad (1)$$

Network parameters were initialized at random in the range [-0.1,0.1] and loss was minimized by stochastic gradient descent with adagrad updates (Duchi et al., 2011).

## 3.2 RECURRENT DFAEs FOR SEQUENCES OF FOVEATIONS

The above architecture is useful for studying single foveations, which is the primary focus of this work. However, we remark that it is straightforward to augment DFAEs with recurrent connections to handle a sequence of foveations similar to what has has been done for solving classification tasks with attention (Mnih et al., 2014). First, augment the foveation function $\phi$ to include a locus $\ell$ on which the fovea is centered. Second, a saccade function $s(h_t; W_s)$ predicts such a locus from the DFAE's current hidden states $h_t$, and finally we make the hidden state recurrent via a function $g(h_{t-1}, W_g)$. Putting this all together yields the following architecture:

$$\hat{x}_t = \phi(x_t, \ell_t) \qquad \text{foveate the image at location } \ell$$
$$h_t = f_e(g(h_{t-1}; W_g), \hat{x}_t; W) \qquad \text{encode: compute new hidden states}$$
$$\ell_t = s(h_t; W_s) \qquad \text{compute new locus for next foveation}$$
$$y_t = f_d(h_t) \qquad \text{decode: reconstruct high detail image}$$

Now the DFAE can handle a sequence of foveations, making it to possible to study the effects of overt attention and also explore the ability to learn from a sequence of foveations (like a language model) rather than a ground-truth image (the analog of which is not available to a human). However, interactions between attention and input acuity are beyond the scope of this work.

# 4 EXPERIMENTS

We are interested in the question of whether an artificial neural network can *perceive* an image from foveated input images and we evaluate this ability by measuring how well the autoencoders can reconstruct a full-resolution image from the low-detail foveated input. In these experiments, we fix the architecture of our network to the family described in the previous section and vary the type of foveation, the number of hidden units and the number of layers and study the learnt features and reconstruction accuracy. We address the following questions: Can the network perceive aspects of the image that are not present in the input? What can it perceive and under what conditions? Can the network perceive color in the periphery? How much capacity is required to perceive missing details? What is the nature of network's learnt solution to the problem of filling-in?

## 4.1 FOVEATION FUNCTIONS

In our experiments, we study several different foveation functions. In many cases, downsampling is employed as part of the foveation function for which we employ the nearest neighbor interpolation algorithm. We chose nearest neighbor because it is especially brutal in that it does not smooth over neighboring pixels when interpolating. Foveation functions include:

- **downsampled factor** $d$ **(DS-D):** no fovea is present, the entire image is uniformly downsampled by a factor of $d$ using the nearest neighbor interpolation method. For example, a factor of 4 transforms a 28x28 image to a 7x7 image and approximately 94% of the pixels are removed. For color images we downsample each channel (RGB) separately, resulting in color distortion. We test factors of 2, 4, 7 (MNIST) and 2, 4, 8 (CIFAR).
- **scotoma** $r$ **(SCT-R):** entire regions ($r = 25\%, 50\%$ and $75\%$) of the image are removed (by setting the intensities to 0) to create a blind spot/region, but the rest of the image remains at the original resolution. We experiment with the location of the scotoma (centered or not).

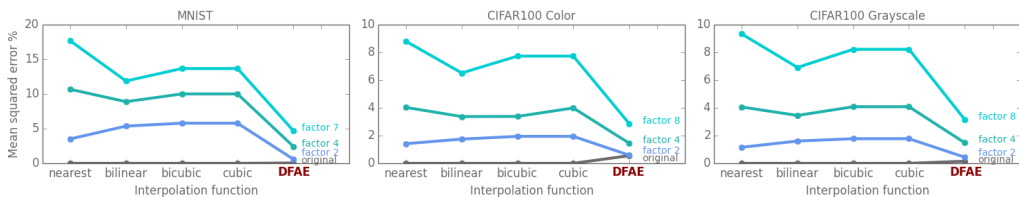

Figure 1: Baseline comparison for downsampled input (DS-D): DFAE on MNIST (resp. CIFAR100 grayscale, CIFAR100 color) with 800 hidden units (resp. 1100, 3100 hidden units).

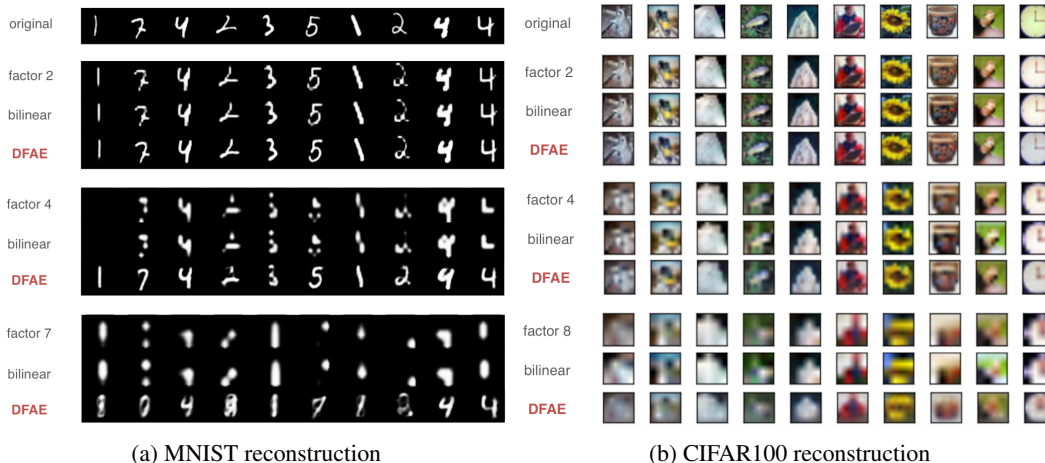

(a) MNIST reconstruction

(b) CIFAR100 reconstruction

Figure 2: Downsampled (DS-D) MNIST and CIFAR100 images reconstruction by DFAE. The top row shows the original image. Each row shows the downsampled input that was used during training, followed by reconstruction images by the bilinear algorithm and DFAEs.

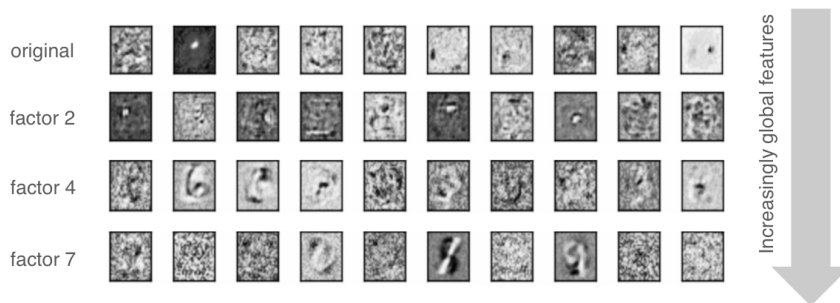

Figure 3: DFAE with 800 units on downsampled (DS-D) MNIST: features become increasingly global as downsampling factor increases.

- **fovea** $r$ **(FOV-R):** only a small fovea $r$ of high resolution ($r = 25\%$ or $6\%$); the rest of the image is downsampled by a factor of $4$.
- **achromatic** $r$ **(ACH-R):** only a region of size $r$ has color; color is removed from the periphery by averaging the RGB channels into a single grayscale channel.
- **fovea-achromatic** $r$ **(FOVA-R):** combines the fovea $r$ with the achromatic region: only the foveated region is in color, the rest of the image is in grayscale and downsampled by a factor of $4$.[1]

---

[1]Note that the achromatic periphery we study here is a more severe distortion than the human periphery, which has dichromatic color reception; though this varies from one individual to another.

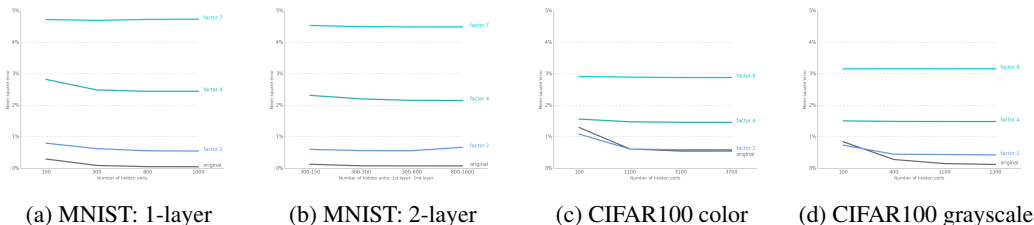

(a) MNIST: 1-layer  (b) MNIST: 2-layer  (c) CIFAR100 color  (d) CIFAR100 grayscale

Figure 4: Error rates of the MLP DFAE are not effected much by network depth or input color.

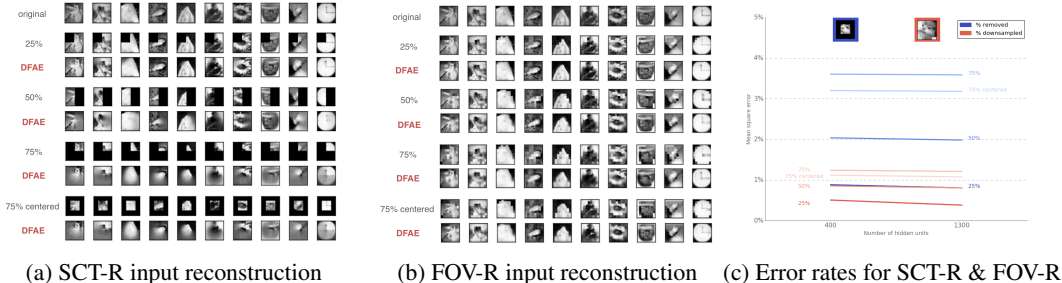

(a) SCT-R input reconstruction  (b) FOV-R input reconstruction  (c) Error rates for SCT-R & FOV-R

Figure 5: Reconstruction examples and errors rates of 1-layer DFAE with foveated input types

## 4.2 DATASETS AND PRE-PROCESSING

We used two datasets in our experiments: MNIST and CIFAR100. The MNIST database consists of 28 x 28 handwritten digits and has a training set of 60,000 examples and a test set of 10,000 examples. Therefore each class has 6000 examples. The CIFAR100 dataset consists of 32 x 32 color images of 100 classes. Some examples of classes are: flowers, large natural outdoor scenes, insects, people, vehicles etc. Each class has 600 examples. The training set consists of 50,000 images and the test set consists of 10,000 images. We trained DFAEs on the MNIST and CIFAR100 dataset (in grayscale and color). We normalized the datasets so that the pixel values are between 0 and 1 and additionally, zero-centered them. This step corresponds to local brightness and contrast normalization. Aside from this, we do no other preprocessing such as patch extraction or whitening.

## 4.3 THE EFFECT OF INPUT ACUITY ON THE NETWORK

The purpose of this experiment is to study how various levels of input acuity affect the performance of the network for the case in which no fovea is available. In addition to input acuity (downsampling factors of 1,2,4,7,8), the other variables to consider are the number of hidden units per layer and the number of layers. In pilot experiments we determined that when the number of hidden units was less than the downsampled input size, DFAEs performed poorly; further, that —unlike CNNs for image classification—additional hidden layers (beyond two) did not improve performance much. Therefore, we report results with overcomplete representations of just one or two hidden layers. First, to establish baselines, we compare one of the networks to various interpolation-based upsampling methods available in image editing software (nearest-neighbor, bilinear, bicubic). The single layer DFAE outperforms these standard algorithms on both datasets (see Figure 1).

For qualitative evaluation, Figure 2 contains examples of the reconstructed images by a single layer DFAE. The images produced by the DFAE is compared to upsampled reconstructions via bilinear interpolation. Especially on MNIST, DFAEs can correctly extract the contents of a downsampled input even when 94% of pixels are removed. A compelling example is that even when faced with a blank input (due to aggressive nearest-neighbor downsampling), as seen in Figure 2a, the DFAE can correctly perceive the digit $1$. Though in general, the performance of DFAEs suffered when the input was downsampled beyond a factor of $4$. When the DFAE made predictions based on the extremely impoverished input, most of the reconstructions were incorrect. Yet, interestingly, the incorrect reconstructions still resembled digits. We hypothesize that this is due to the global feature functions

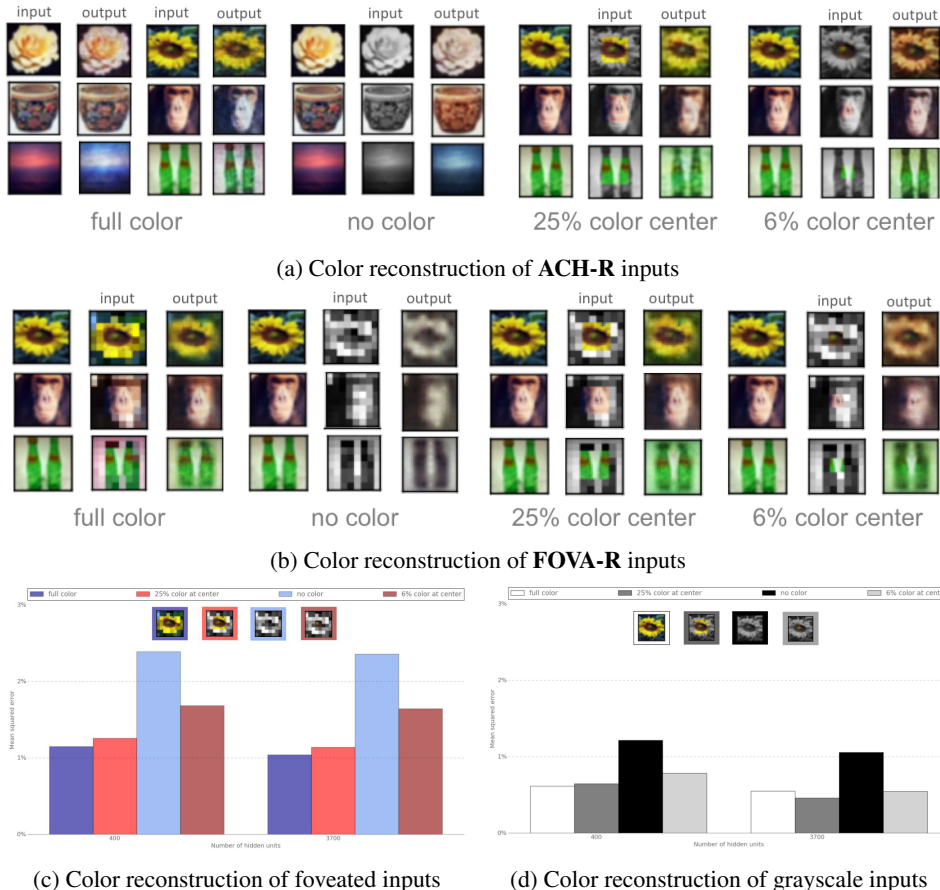

Figure 6: Color reconstruction examples and errors rates of 1-layer DFAE

learnt by the network. Indeed, we observe in Figure 3 that for MNIST, as the downsampling factor increases, the global structure in the features[2] also increases: when the input was downsampled by a factor of two (resp. four, seven), it was forced to learn stroke like features (resp. full/partial digits, superimposed digits). A curiously similar result was observed by Vincent et al. Vincent et al. (2010), where their denoising autoencoder learnt global structures when it was trained on randomly noisy inputs.

On the other hand, many filters learnt on CIFAR100 images were not directly interpretable. Though, in some cases the network learnt global features such as a specific color gradients or locally circular blobs which probably enabled it to be better at reconstructing low frequency shape information, color gist, and landscapes particularly well. The reconstructed natural images as seen in Figure 2b show that the DFAE learnt a smoothing and centering function. The DFAEs employed here could predict the shape of objects in the natural images but not the high frequency details (e.g., texture).

## 4.4 RECONSTRUCTING FOVEATED INPUTS

Until now, we evaluated DFAEs on uniformly downsampled images, but we are especially interested in the case for which some high resolution input is available, more closely resembling the retina. In this section, we evaluate DFAEs on foveated inputs, **SCT-R** and **FOV-R**, as described in Section 4.1.

The scotoma allows us to isolate the contribution of the fovea alone (without help from the periphery). Variable-sized areas of region ($r = 25\%$, $50\%$, $75\%$, $75\%$ centered) were removed from the original input. The location of removal was chosen randomly from the four quadrants of the input image, except for the condition where 75% of the image around the center was removed. Since

---

[2]each feature is the the final output layer weights corresponding to a particular hidden unit

a majority of the input images have a subject of interest, we tested if the central region contained enough information to reconstruct the rest of the image.

The reconstructions in Figure 5a show the DFAE does not perform well when $r > 25\%$. When $r = 50\%$, the DFAE can only reconstruct landscapes, shape and symmetry, demonstrating its ability to extract low frequency information. When $r = 75\%$ and 75% centered, the reconstruction process breaks down and the DFAE cannot predict the input beyond the given region of information. The filters learnt under these conditions look similar to the downsampled condition, but with larger blobs.

In **FOV-R** inputs, $r$ is the same as **SCT-R** inputs and we chose to use downsampling factor 4 for regions outside the fovea since previous experiments revealed that DFAEs cannot reconstruct inputs downsampled beyond this factor. Figure 5b shows the reconstructed images from **FOV-R** inputs and Figure 5c show the error rate of reconstruction. The cluster of red lines with lower error rates show that the DFAE performed considerably well with **FOV-R** than **SCT-R** inputs. The performance was better ( 1% error for $r = 75\%$ centered) than an DFAE trained with uniformly downsampled inputs (1.5% error). This result is not surprising, given that **FOV-R** contains additional information from regions outside the fovea. These results suggests that a small number of foveations containing rich details might be all these neural networks need to extract contents of the input in higher detail.

## 4.5 Reconstructing color from foveated inputs

It is well known that the human visual system loses chromatic sensitivity towards the periphery of the retina. Recently, there has been interest in how deep networks, specifically convolutional neural networks (CNNs), can learn to color grayscale images  (Dahl) and learn artistic style Gatys et al. (2015). Specifically in Dahl reconstructions from grayscale images, numerous cases of the colorized images produced were muted or sepia colored. The problem of colorization which is inherently ill-posed was treated as a classification task in these studies. Can DFAEs perceive color if it is absent in the input?

We investigated this question using **ACH-R** and **FOVA-R** inputs described in section 5.1. The regions of color tested were $r = 0\%$ or no color, 6%, 25% and 100% or full color. Figure 6a and 6b show examples of color reconstructions of the these input types. When the DFAE is trained with full color **ACH-R** inputs, it can make mistakes in reconstructing the right color as seen in Figure 6a. For example: it colors the yellow flower as pink and the purplish-red landscape as blue. When the input is grayscale (no color, $r = 0\%$), the colorizations are gray, muted, sepia toned or simply incorrect in the case of landscapes. But if there is a fovea of color, the single layer DFAE can reconstruct the colorizations correctly. Of course, if the fovea of color is reduced, i.e. 6%, the color reconstruction accuracy falls off but not too drastically. For example, it predicts a yellowish tone for the sunflower among a bed of brown leaves. The critical result is that the performance difference between 100% or full colored inputs and foveated color inputs is small as seen in Figure 6c and 6d. These results suggest that color reconstructions can be just as accurate if these networks can figure out the color of one region of the image accurately as opposed to every region in the image. Similar to the human visual system, these networks are capable of determining accurate colors in the periphery if color information is available at foveation.

## 5 Discussion and Conclusions

We presented a framework for studying whether a given neural network architecture can perceive visual details not explicitly present in the input. Using the framework, we studied a simple fully-connected network and found it could fill in missing details such as shape, color and contrast, but not texture. We found that the network compensates for missing details by learning global features which when appropriately activated by the surround, effectively infer the missing information. Thus, the network performs perceptual filling-in even though the layers lack the lateral connections and grid-like structure that have been postulated in the isomorphic filling in theory  (von der Heydt et al., 2003). Further, the network appears to learn correlations between color and shape: when presented a sunflower input in which only the dark brown center and the inner yellow petals were in color, the network correctly perceived that the surrounding leaves are green; in the case of landscapes, the network incorrectly perceived the sky as blue, even when a small fovea of red sunset is present in the input. Upon inspection, the network indeed learns global color features and these features may

be activated by input patterns corresponding to certain shapes causing the network to occasionally over-generalize, but further investigation is needed to fully confirm this hypothesis.

A potentially interesting line of future research would be to employ the DFAE framework to test the existing hypotheses for how our own visual system performs *perceptual filling in*: e.g., the process by which our brain infers information that is not explicitly present in our sensory input from the surround (Komatsu, 2006). The specific architectures we studied here are a far cry from the true mechanisms, but in future work we could implement several architectures that capture the essence of competing theories: Which neural architectures exhibit similar behaviors and abilities as humans? Will the architecture be fooled in the same way as humans by optical illusions such as the *Cornsweet illusion* or the *Troxler effect?* When does it perceive "The Dress" as blue and gold or white and black? By answering questions such as these, the DFAE framework could provide evidence for one theory (or neural mechanism) over another, and these theories can be further investigated with behavioral and neurological experiments.

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
