# Peer review of "Filling in the details: Perceiving from low fidelity visual input"

_ICLR 2017 — rejected_

[Official Review · AnonReviewer2 · rating 6 · confidence 4 · 16 Dec 2016]
**Interesting idea, evaluation could be improved**

I like the idea the paper is exploring. Nevertheless I see some issues with the analysis:

- To get a better understanding of the quality of the results, I think at least some state-of-the-art comparisons should be included (e.g. by setting d times d pixel patches too their average and applying a denoising autoencoder). If they perform significantly better, then this indicates that the presented model is not yet taking all the information from the input image that could be used.
- SCT-R and FOV-R are supposed to test how much information can be restored from the Fovea alone as opposed to the Fovea together with low resolution periphery. However, there is an additional difference between the two conditions: According to the paper, in SCT-R, part of the image was set to zero, while in FOV-R it was removed alltogether. With only one or two hidden layers, I could easily imagine this making a difference.
- On page 4, you compare the performance of FOV-R (1% error) with that of DS-D (1.5%) and attribute this to information about the periphery that the autoencoder extracts from the fovea. While this might be the case, at least part of the reduced error will be due to the fact that the fovea is (hopefully) perfectly reconstructed. To answer the actual question "how much additional information about the periphery can be extracted from the fovea", you should consider calculating the error only in the periphery, i.e. the part of the image where DS-D and FOV-R got exactly the same input for. Then any decreased error is only due to the additional fovea information.

Other issues:
- The images in Figure 2 (a) and (b) in the rows "factor 2", "factor 4", "factor 8" look very blurry. There seems some interpolation to be going on (although slighly different than the bilinear interpolation). This makes it hard to asses how much information is in these images. I think it would be much more insightfull to print them with "nearest" interpolation.
- Figure 3 caption too vague. Maybe add something like footnote 2?
- Often figures appear too early in paper which leads to lots of distance between text and figures.

[Official Review · AnonReviewer3 · rating 5 · confidence 5 · 16 Dec 2016]
**This paper is well motivated**

This paper is motivated by the ability that human's visual system can recognize contents of environment by from critical features, and tried to investigate whether neural networks can also have this kind of ability.  Specifically, the paper proposed to use Auto-Encoder (AE) as the network to reconstruct the low fidelity of visual input. Moreover, similar to Mnih et al. (2014),  the paper also proposed to use a recurrent fashion to mimic the sequential behavior the  human visual system. 

I think the paper is well motivated. However, there are several concerns:
1. The baselines of the paper are too weak. Nearest neighbor, bilinear, bicubic and cubic interpolations without any learning procedure are of course performed worse than AE based models. The author should compare with the STOA methods such as

[Official Review · AnonReviewer1 · rating 4 · confidence 3 · 18 Dec 2016]
**some interesting cases, but lacks focus**

This paper aims to characterize the perceptual ability of a neural network under different input conditions.  This is done by manipulating the input image x in various ways (e.g. downsamplig, foveating), and training an auto-encoder to reconstruct the original full-resolution image.  MSE and qualitative results are shown and compared for the different input conditions.

Unfortunately, this paper seems to lack focus, presenting a set of preliminary inspections with few concrete conclusions.  For example, at the end of sec 4.4, "This result is not surprising, given that FOV-R contains additional information .... These results suggests that a small number of foveations containing rich details might be all these neural networks need....".  But this hypothesis is left dangling:  What detailed regions are needed, and from where?  For what sort of tasks?

Secondly, it isn't clear to me what reconstruction behaviors are caused by a fundamental perception of the input, and what are artifacts of the autoencoder and pixelwise l2 loss?  A prime example is texture, which the autoencoder fails to recover.  But with a pixelwise loss, the network must predict high-frequency textures nearly pixel-for-pixel at training time; if this is impossible, then it will generate a pixelwise average of the training samples --- a flat region.  So then the network's inability to reconstruct textures is due to a problem generating them, specifically averaging from the training loss, not necessarily an issue in perceiving textures.  A network trained a different way (perhaps an adversarial network) may infer a texture is there, even if it wouldn't be able to generate it in a pixelwise l2 sense.

Similarly, the ability to perform color reconstruction given a color glimpse I think has much to do with disambiguating the color of an object/scene:  If there is an ambiguity, the network won't know which to "choose" (white flower or yellow flower?) and output an average, which is why there are so many sepia tones.  However, in its section on this, the paper only measures the reconstruction error for different amounts of color given, and does not drill very far into any hypotheses for why this behavior occurs.

There are some interesting measurements here, such as the amount of color needed in the foveation to reconstruct a color image, and the discussion on global features, which may start to get at a mechanism by which glimpses may propagate to an entire reconstruction.  But overall it's hard to know what to take away from this paper.  What are larger concrete conclusions that can be garnered from the details, and what mechanisms bring them about?  Can these be more thoroughly explored with more focus?

[Final Decision · Program Chairs · 06 Feb 2017]
**ICLR committee final decision**

The program committee appreciates the authors' response to concerns raised in the reviews. Unfortunately, reviews are not leaning sufficiently towards acceptance. Reviewers find this direction of exploration to be interesting, but a bit preliminary at the moment. Authors are strongly encouraged to incorporate reviewer comments to make future iterations of the work stronger.